# Soil Management and Microclimate Effects on Ecosystem Evapotranspiration of Winter Wheat–Soybean Cropping in Northern Alabama

**Maheteme Gebremedhin ***[ID]**, Jacob Brown and Ian Ries**

College of Agriculture, Communities and the Environment, Kentucky State University, 400 East Main St., Frankfort, KY 40601, USA
* Correspondence: maheteme.gebremedhin@kysu.edu

**Abstract:** Nearly all the current winter wheat–soybean cropping systems occurring in the southeastern United States (SE) region are rainfed, as such, precipitation (*P*) underpins energy partitioning. We investigated, using the eddy covariance technique, the seasonal and interannual variability and rate and trend of energy partitioning, i.e., sensible and evapotranspiration (*ET*), for rainfed soybean and winter wheat cover cropping at Winfred Thomas Agricultural Research Station (WTARS) in Hazel Green, Alabama. Yearlong cumulative *ET* of 493, 743, and 746 mm during 2007, 2008, and 2009 relative to cumulative precipitation of 567, 1280, and 1356 mm y$^{-1}$ resulted in a higher mean *ET/P* ratio of 0.87, in 2007, medium 0.58 in 2008, and lowest 0.55 in 2009. Mean daily *ET* for the cover crop and soybean ranged between 1.70 and 2.44 mm d$^{-1}$ and 1.82 to 2.83 mm d$^{-1}$, respectively. Overall, our findings suggest total and seasonal precipitation distribution were major controlling factors in the partitioning of the energy and water budgets. This study provides evidence that changes in rainfall frequency and intensity in the SE will likely alter the regional croplands hydrology with implications on water resource management decisions in rainfed agriculture.

**Keywords:** eddy covariance; energy partitioning; energy closure; sensible heat; latent heat; ground heat; evapotranspiration

## 1. Introduction

Much of the cropland across southeastern United States (US) is rainfed; as such precipitation governs nearly all processes of the terrestrial hydrologic cycle, including evapotranspiration, (*ET*), runoff, and soil moisture availability. In the southeast (SE), full-season double-cropped soybean (*Glycine max* (L.) Merr.) following winter wheat (*Triticum aestivum* L.) is a common cropping system. Of particular significance, the soybean–wheat cropping system has a dual-added economic benefit and sustainability appeal [1,2]. Of the total 3.5 million hectares of cropland in the US, nearly a third (~1.09 million hectares) of the southeast cropland is double cropped [3]. For example, in a span of two years, in 2008, there was ~4.4 million hectares of land used in double cropping, but that figure dropped by nearly half (to 2.2 million hectares) in 2010 [4,5]. However, the recent gradual increase in cover crops used in many parts of SE and across the US is prompting studies on their potential impact on surface energy budget and on the region's hydrologic cycle. Additionally, croplands of the SE, could potentially be vulnerable to a shifting and warming climate in the region marked by strong seasonal and interannual variability in precipitation [6]. In this humid subtropical climate, impacts of changing seasonal and year-to-year variability (duration, intensity, and distribution) in precipitation plays a critical role in modulating surface radiation budget and has a profound influence on regional energy cycle [7,8]. The croplands of the region play an important role in regulating the regional hydrologic balance due to their widespread distribution and ability to exchange a significant amount of water vapor with the atmosphere [9]. Thus, cropland water management and policy issues need

to be addressed that account for the hydrologic sustainability of the region [10]. The need for region-wide networked observing system capabilities is rarely justified for croplands among the flux community [11]. In part, this is because of the diversity of cropping practices and soil management approaches employed over these fragmented landscapes. Recently, however, the widespread use of flux towers (e.g., eddy covariance technique) over managed agroecosystems is increasingly used in monitoring temporal variability in surface energy budgets across the world [12]. Such networked observation systems, combined with other efforts to assess the methods and approaches of assessing energy balance and ET of croplands, is likely to advance our understanding of drought (e.g., the drought conditions experienced during 2005/2006/2007 and summer 2012/2013 growing seasons) on surface energy partitioning [13].

The predominantly rainfed croplands in the SE occupy $53.9 \times 10^6$ ha, nearly 4% of cropland, the second largest agricultural region in the North American continent [14] yet their roles in influencing the regional hydrologic cycle is poorly understood. Nearly all the current double winter wheat–soybean cropping systems occurring in the southeastern SE region are rainfed, and, as such, precipitation ($P$) underpins energy partitioning. Because of the strong dependency on rainfall, the net hydrologic balance between precipitation and $ET$ is likely to be influenced by factors that underpin climate patterns across the region. In this humid subtropical climate zone, impacts of year-to-year variability in precipitation exert considerable influence on regional hydrology and productivity [15,16]. Despite the expansion (given its geographical expanse) of the cropping system, there is a lack of concurrent growth in the understanding of ecological processes and the how surface energy is partitioned into sensible ($H$), latent ($LE$), and ground ($G$) heat fluxes [17,18]. In recent years, however, the need for long-term measurement for assessing and quantitatively evaluating the energy and water flux of croplands, in a consistent and long-term basis, is being recognized [11].

In recent years, SE ecosystems have experienced broad-scale patterns of temperature and precipitation variability, resulting in significant crop loss, decreased reservoir levels, and conflicts over water resources among neighboring states [19]. For these reasons, functional responses of croplands to microclimate and crop management could have profound impacts on the influence of water and energy exchanges, crop sustenance, and productivity on the regional climate [20,21]. To assess the impact of enhanced drought on surface energy balance, we focused on the period from 2007 to 2009. The year 2007 was arguably the most notable drought year interspaced by intermediate and above normal 2008 and 2009 years, in terms of precipitation, respectively. The dissimilarity in amount and distribution of rainfall, among the years, allowed us to robustly quantify the variation and understand how precipitation and temperature impact the surface energy budget of on cropped lands, specifically ET. Previous studies have shown that agricultural productivity and surface energy balance may be sensitive to year-to-year variation in precipitation and warming [22,23]. Thus, providing a robust estimate of the amount of surface energy exchange provides a base to predict the effects of future cropland expansion on the regions' net hydrologic balance, to resolve regional water budget, and to guide the development of sector-based, sustainable water resource management and policy [7,19].

In this study, we seek to estimate how site microclimate and cropping practices influence seasonal energy cycling and $ET$ exchange over a rainfed cropping system in the SE. Data collected from a worldwide network of flux measurement stations have enabled scientists to assess the sensitivity of hydrologic processes to environmental factors and to quantify the magnitude of $ET$ fluxes in several ecosystems, including croplands, grasslands, and forests, at spatial scales ranging from plot to regional to continental [24]. We examined the temporal patterns of energy partitioning and $ET$ of the winter wheat–soybean double cropping system. Broadly, our goal was to better understand the biophysical factors that regulate energy balance and $ET$ of this cropping system under a humid northern Alabama climate. The objectives of the study were: (i) to quantify the water and energy (surface energy budget) and gain better understanding of biophysical controlling factors (phenology,

precipitation, and temperature), (ii) examine the functional relationship between ET partitioning and soil moisture availability, (iii) provide a quantitative assessment of seasonal and inter-annual variation in energy partitioning and *ET*, and (iv) better understand the functional relationships between energy flux and underlying biophysical factors across temporal scales. Specific questions we will be addressing are:

What biophysical factors (e.g., moisture, light, temperature) drive surface–atmosphere fluxes over a typical cropland? How do croplands regulate the balance of LE and H as they grow/mature? How are impacts of seasonal droughts manifested on soybeans and cover crops, and what is the average soil and plant moisture threshold (pressure) for continued transportation?

## 2. Materials and Methods

### 2.1. Micrometeorological Measurements

The Winfred Thomas Agricultural Research Station (WTARS) is a 40 ha agricultural research farmland in Hazel Green, Alabama (34°53 N, 86°34 W, 191 m above mean sea level), Alabama Agricultural and Mechanical University (AAMU). Situated in the Cumberland Plateau, the climate at WTARS is humid subtropical with an average annual temperature of 15.8 °C and total annual precipitation 1460 mm (NOAA, National Climatic Data Center, https://www.ncei.noaa.gov/hun/?=climate_normals_1981-2010_huntsville (accessed on 13 August 2022)). Detailed site description can be found elsewhere [24,25].

The EC flux measurement tower is located on a slightly eastward slope (<0.5%) with a sufficiently wide and horizontal fetch of at least 200–300 m in the major wind directions. The EC tower was instrumented with a fast response 3D sonic anemometer (CSAT-3, Campbell Scientific Inc. Logan, UT, USA) that measures three-dimensional wind speed (m s$^{-1}$) in space and sonic air temperature ($T_s$, °C). Vertical exchange of $H_2O$ vapor, sensible heat flux, and latent heat flux were measured using the eddy-covariance technique. Water vapor concentration was measured, in situ, using an open-path infrared gas analyzer (IRGA, Li7500 LI-COR Inc., Lincoln, NE, USA). Both the CSAT and the IRGA were collocated at the end of a 0.82 m boom at a height of 3 m above ground level. The two sensors, operated at 10 Hz frequency, sampled the turbulence and data was collected and stored in a datalogger (CR5000, Campbell Scientific Inc., Logan, UT, USA). The IRGA was slightly tilted (~15° from the vertical axis of the IRGA) to prevent rainwater accumulation and dew deposition on the measuring window and is displaced 0.15 m behind the CSAT-3, facing the dominant prevailing winds from the southeast. The IRGAs were calibrated using a dew point generator (LI-610, Li-Cor, Lincoln, NE, USA) and high purity $CO_2$ gas (Airgas, Huntsville, AL, USA) and dry $CO_2$ free gas, for spanning and zeroing, respectively. Calibrations were performed every 3 to 6 months following AmeriFlux protocols [26,27].

### 2.2. Data Processing

The raw flux data was processed using Edire (Micrometeorological Data processing Software, Institute of Atmospheric and Environmental Science, School of GeoSciences, University of Edinburgh, http://www.geos.ed.ac.uk/abs/research/micromet/EdiRe, (accessed 20 June 2007)).

Missing flux data were primarily due to low-wind turbulence, system downtime during calibration, inclement weather, and sensor malfunctioning. First, data were removed when measured values were made under insufficient turbulent mixing or in a stable atmosphere at night using a threshold friction velocity ($u$ * of 0.1 m s$^{-1}$) [28]. In addition, data were removed from the half-hourly averages if a data point falls within one of the following rejection criteria: (i) during rain events or morning dew, (ii) for incomplete 30 min data (iii) when variance tests indicated values above an established threshold value (e.g., $\sigma^2_{CO2} < 75$) (data not shown), (iv) when plausibility tests indicated calculated flux values were out of range (i.e., ± 3 SD), (v) both daytime and nighttime negative fluxes of water vapor were set to zero. The total remaining data coverage, on average, was ~78% for the three years (2007–2009) of data.

The buoyancy (stability) parameter $\xi = \frac{z}{L}$ (nondimensional), was used to define three stability classes, namely unstable $\xi < -0.1$, neutral $-0.1 < \xi < 0.1$, and stable $\xi > 0.1$ atmospheres, where $z$ is the measurement height (m) and $L$ is the Monin–Obukhov buoyancy length (m) scale expressed as,

$$L = -\frac{u_*{}^3}{\left[k\left(\frac{g}{T_0}\right)\left(\frac{H_0}{pc_P}\right)\right]} \tag{1}$$

where, $u_*$ is the friction velocity (m s$^{-1}$, expressed as $u_* = \left|\left(\sqrt{u'w'}\right)\right|^{0.25}$, $k$ is the von.

Karman constant (=0.41), $u'$ and $w'$ are the fluctuating horizontal and vertical wind speeds, respectively, from the 30 min average wind speed, $g$ is acceleration due to gravity (9.81 m s$^{-1}$), $T_0$ surface air temperature in $_0$K, $H_0$ is surface sensible heat flux, (W m$^{-2}$), $p$ is air density (kg m$^{-3}$), and $C_p$ is the specific heat capacity of air, kJ/kg $_0$K.

The net all wave radiation is the sum of all incoming and outgoing radiant energy flux components,

$$R_n = (S_d - S_u) + (L_d - L_u) \tag{2}$$

where $R_n$ is the net radiation, $S_d$ and $S_u$ are the incoming short-wave radiation and reflected short wave radiation, respectively, $L_d$ and $L_u$ are the downward and outgoing thermal long wave radiation, respectively. Positive values of all other terms in the equation mean a loss of energy from the surface to the atmosphere. The turbulent sensible ($H$) and latent ($LE$) heat fluxes were computed as follows:

$$H = \rho_a C_P \overline{w'T'}_s \tag{3}$$

where $\rho_a$ is dry air density (Kg m$^{-3}$), $C_p$ is the specific heat capacity of the air at constant pressure (1004 J kg$^{-1}$K$^{-1}$). The overbar indicates the averaging period, in this case 30 min. Similarly, $LE$ (W$^{-2}$) was calculated as:

$$LE = \lambda \overline{w'q'} \tag{4}$$

where $\lambda$ is the latent heat of vaporization (=2.45 MJ kg$^{-1}$) and $q'$ is the fluctuation about the mean of density of water vapor (Kg m$^{-3}$), $\rho_w$, in air (=1000 Kg m$^{-3}$). Evapotranspiration ($ET$, mm) was estimated using the relationship:

$$ET = \frac{LE}{\rho_w \lambda} \tag{5}$$

Daily, seasonal, and annual $LE$ totals were calculated by integrating the 30 min observations. The energy balance at a surface can be thought as an accounting of energy gains and losses within a specified time interval (e.g., 30 min, daily monthly seasonally and annually). We assessed the relative energy balance closure by plotting the sum of the $H$ and $LE$ against the available energy (i.e., $R_n$–$G$) and the mathematical expression for a complete energy closure assumes:

$$y + a(R_n - G) = H + LE + G + \varepsilon \tag{6}$$

where $a$ and $y$ are the slope and intercept of the linear regression, respectively, and $G$ is ground heat flux. Units for $R_n$, $G$, $LE$, and $H$ terms are W m$^{-2}$. The last term, $\varepsilon$, is the error term, which includes both instruments and random error. $G$ was directly measured using a soil heat plate at 10 cm depth. The partitioning of $R_n$ into $H$ and $LE$ was explored using the Bowen ratio ($\beta$):

$$\beta = \frac{H}{LE} \tag{7}$$

Changes in surface reflectivity (albedo, $\alpha$) across growing and non-growing seasons was calculated as the ratio of the reflected to the incoming short-wave radiation,

$$\alpha = \frac{S_u}{S_d} \tag{8}$$

To avoid the compounding effect of solar angle on crop $\alpha$, daily and seasonal trends were only analyzed for specific hours, i.e., between 10:00 h and 14:00 h local standard time (LST).

### 2.3. Supporting Micrometeorological Measurements

Supporting site climate and soil data were collected using standard micrometeorological instrumentation. Precipitation was recorded using a tipping bucket rain gauge (TE525-LC, Texas Electronics, Campbell Scientific Inc., Logan, USA). Air temperature ($T_a$, °C) and relative humidity ($RH$, %) measurements were made at 2.5 m above ground level (HMP 45, Viasala, Helsinki, Finland). Downward and reflected short wave and incoming and emitted long-wave radiation were measured using a pair of pyranometers and pyrgeometers, respectively (CNR-1, Kipp and Zonen, Delft, The Netherlands), mounted on a horizontal boom 2.45 m above ground. Soil volumetric water content ($SWC$) was measured with 3 time-domain reflectometers (CS-616, Campbell Scientific Inc., Logan, UT, USA) and three soil heat flux plates (HFP01, Hukseflux, Delft, The Netherlands) were installed at 0.10, 0.15, and 0.40 m below the surface.

### 2.4. Tillage and Cropping Management

The cropping system at our site is a single main crop (soybean) with a winter wheat cover crop. Typically, the cover crop–soybean cropping seasons in north Alabama are interceded by two short fallow periods (defined as a period of bare ground for a period of 2–6 weeks). Each occur prior to the cover crop and soybean planting (i.e., October, winter fallow and between mid-May and early June, spring fallow). Beginning in fall of 2006 and subsequent winter growing seasons, the site was cover cropped with winter wheat (*Tricticum aestivum* (L.)), from November through May. The cover crop is usually cut green and was usually left on the ground as green manure. The main summer season crop, roundup ready soybean (*Glycine max* (L.), (Monsanto Co., St. Louis, MO, USA) was planted in June and harvested in October.

The soil at WTARS is in the Abernathy–Decatur soil series (fine-silty, siliceous, active, thermic Fluventic Humic Dystrudepts), commonly found on flat terrain (0 to 2% slopes). It is one of the most extensive soil types in northern Alabama along the southern end of the Appalachian Ridges and Valleys (USDA-NRCS, Soil Data mart; http://websoilsurvey/nrcs/usda.gov/app/WebSoilSurvey/aspx, (accessed on 4 August 2022)). The soil profile is characterized by a thin organic layer (1.85% in the upper 10–15 cm deep), slightly acidic in pH, and moderately well drained with high permeability.

In our no-till procedure, seeds were planted below the soil surface, underneath the pre-existing crop residue. The cover crop (winter wheat) was sown at a rate of 180–200 kg seeds/ha. The cover crop was cut when green and left on the ground as plant residue prior to soybean planting. The seeding density for soybean (summer crop) was at ~185–300 kg seeds/ha$^{-1}$ at a row spacing of 0.38 m.

## 3. Results

### 3.1. Site Microclimate

Marked differences in total precipitation among the years characterized the study period, including the notable drought year of 2007, which impacted winter wheat cover crop and soybean growing seasons (Figure 1). Precipitation varied considerably among the years, spanning from the record driest year of 2007 (567 mm) to the relatively wet year of 2009 (1356 mm). Similarly, growing period precipitation varied from a minimum of 112 mm in 2007 to a maximum of 827 mm in 2009. In 2007, excluding August, monthly totals

deviated from long term averages for the entirety of 2007. Based on the 30 yr (1970–2000) average of 1460 mm for the region, the observation years can be classified as dry (2007), normal (2008), and wet (2009). Due to the below average recorded precipitation in the 2007 growing season, the lowest *SWC* observed was the 0.08 $m^3$ $m^{-3}$. High *SWC* in the deeper soil profile was evident, mainly caused from inaccessibility by roots and high soil water retention capacity during the non-growing season.

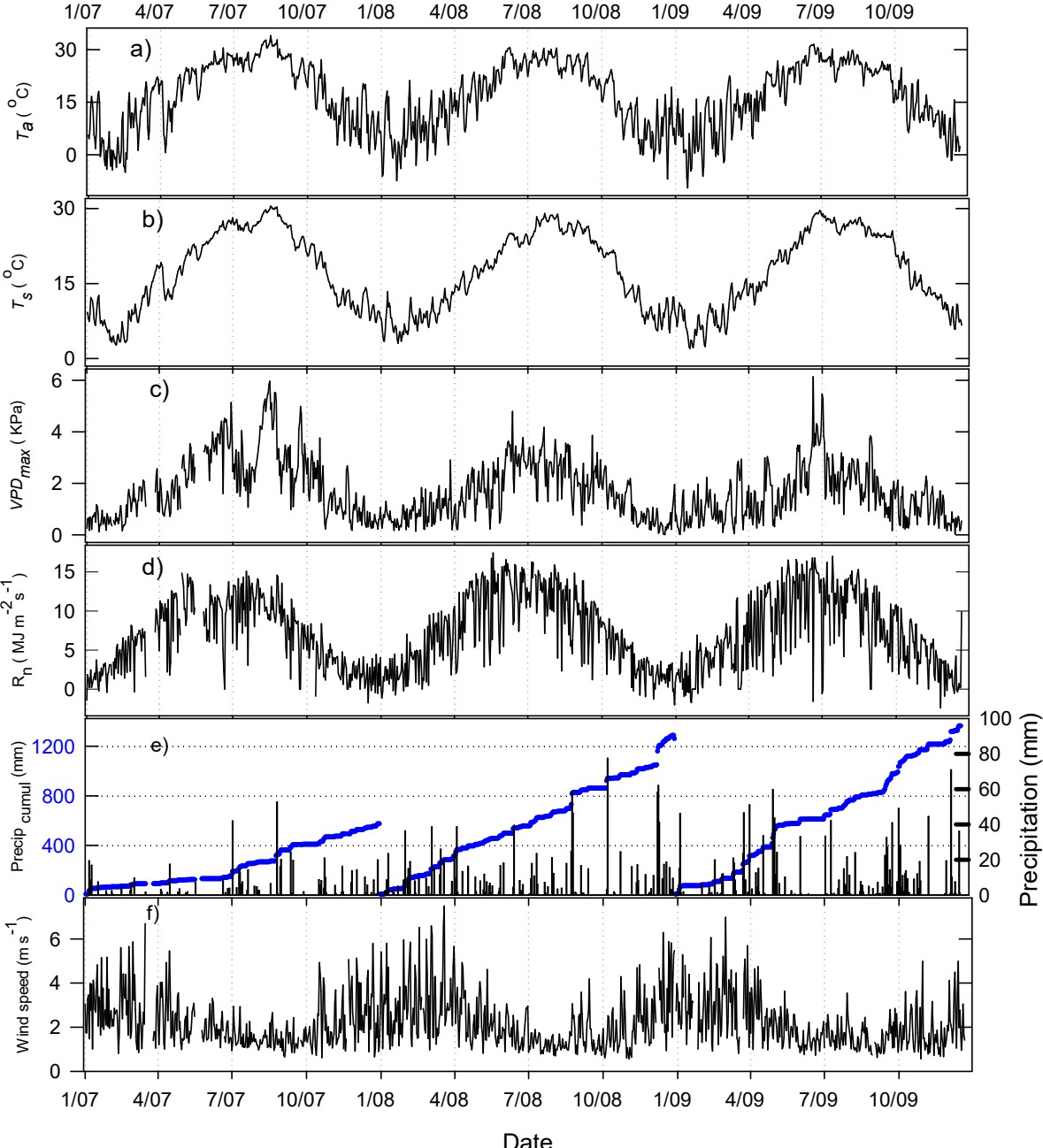

**Figure 1.** Time series of daily average (**a**) mean daily air temperature (sonic, $T_s$), (**b**) mean surface soil temperature (**c**) maximum daily vapor pressure deficit (*VPD*) (**d**) mean net radiation ($R_n$) (**e**) daily and year cumulative precipitation (diagonal line) in (mm), and (**f**) mean daily horizontal wind speed (m s$^{-1}$) at WTARS for the study period 2007–2009.

Compared to the long-term 1970–2000 means, annual average air temperature ($T_a$) at WTARS was slightly warmer, with a notable positive increase (3.5 °C) in the summer (June, July, and August) of 2007. Several days in August 2007 approached daytime maxi-

mum of ~40 °C, making it the warmest month in 150 y record. The 2007 winter growing season (except for February) was also warmer than 2008 and 2009. On average, the first half of 2008 was cooler than the first half of 2009, which was warmer by 1.3 °C. During spring (March–May), the trend of increasing temperature continued, with March being the warmest (3.6 °C higher than the long-term average).

In 2007, monthly mean vapor pressure deficit (*VPD*) ranged from 0.53 to 3.6 kPa and averaged 0.56 kPa, while in the following years it ranged 0.53 to 3.6 kPa. Midday maximum *VPD* exceeded 3.5 kPa on several days in summer months of July and August in 2007 but reached 3.0 kPa in the summer of 2008. During the winter months of the winter wheat growing season, maximum daily *VPD* varied between 0.2 to 2.5 kPa and averaged 1.5, 1.6, and 1.8 kPa in 2007, 2008, and 2009. Maximum daily *VPD* observed was largest in summer months of the soybean growing seasons and varied between 0.2 to 5.5 and averaged 2.6, 3.5, and 2.9 kPa in 2007, 2008, and 2009, respectively.

### 3.2. Diurnal and Seasonal Variation in Net Radiation Components

Across the years, $R_n$ ranged from a monthly minimum of 34 W m$^{-2}$ (January) to a maximum of 243 W m$^{-2}$ (July) with a small year-to-year variation between 303 and 351 W m$^{-2}$, despite differences in the components (Figure 2). The highest annual total $R_n$ of 351 W m$^{-2}$ in 2007 was attributed most likely to more frequent cloudless days, compared to 2008 and 2009, as was evidenced by the lowest annual precipitation in 2007. Strong upward long wave radiation $L_u$ characterized $R_n$, which was the largest fraction of all the components. Daily maximum $L_u$ varied between 493 and 535 W m$^{-2}$ and averaged 297 W m$^{-2}$, while down welling long-wave radiation ($L_d$) remained relatively constant over the study period. The energy partitioning exhibited distinct seasonal patterns (Table 1).

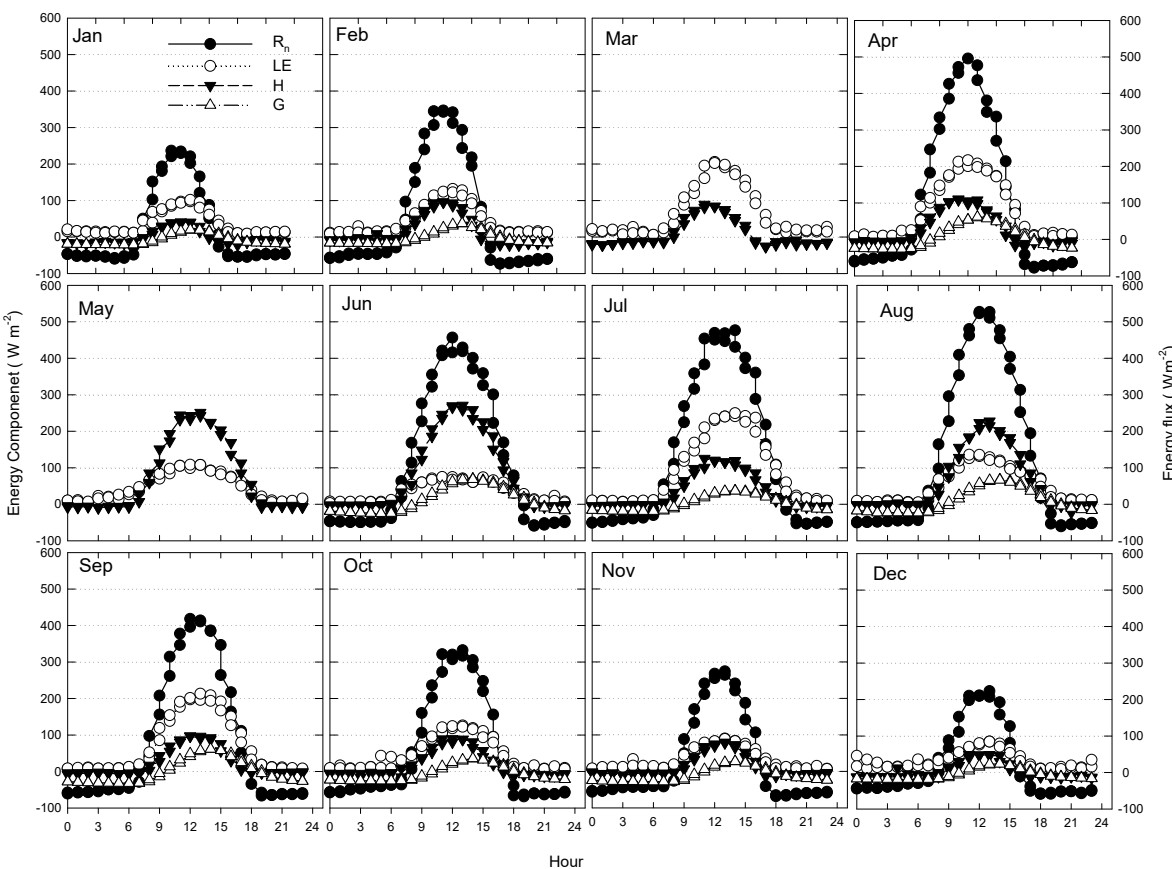

**Figure 2.** Mean monthly diurnal course of individual components of net radiation ($R_n$), latent (*LE*), sensible (*H*), and ground (*G*) heat fluxes in 2007.

**Table 1.** Seasonal energy partitioning of the net radiation ($R_n$) into latent heat ($LE$), sensible heat ($H$), and ground heat flux ($G$) and Bowen ratio ($H/LE$) during growing season of (winter wheat cover crop and soybean) and non-growing seasons. Numbers in bold are annual averages.

| Year | Crop | DOY | $R_n$ | $LE$ | $H$ | $G$ | $H/L$ |
|------|------|-----|-------|------|-----|-----|-------|
| 2007 | Cover crop | 304–130 | 62.9 | 60.5 | 15.7 | 17.5 | 0.26 |
| | Non-growing | 132–152 | 48.7 | 43.0 | 81.8 | 33.6 | 1.90 |
| | Soybean | 182–273 | 117.1 | 87.3 | 41.9 | 30.6 | 0.48 |
| | Cover crop | 275–306 | 122.1 | 49.1 | 21.4 | 13.7 | 0.44 |
| Year total | | | **351.0** | **239.9** | **161.0** | **95.4** | **0.67** |
| 2008 | Cover crop | 304–130 | 57.8 | 61.2 | 10.3 | 15.6 | 0.17 |
| | Non-growing | 132–152 | 54.0 | 77.0 | 72.3 | 41.8 | 0.94 |
| | Soybean | 182–273 | 137.8 | 111.2 | 34.3 | 35.1 | 0.31 |
| | Cover crop | 275–300 | 54.0 | 52.4 | 32.5 | 17.5 | 0.62 |
| Year total | | | **303.6** | **301.8** | **149.0** | **110** | **0.49** |
| 2009 | Cover crop | 304–130 | 67.1 | 89.8 | 10.8 | 17.1 | 0.12 |
| | Non-growing Soybean | 132–152 | 151.9 | 86.7 | 83.4 | 35.3 | 0.96 |
| | Cover crop | 183–326 | 102.7 | 76.2 | 40.0 | 25.2 | 0.52 |
| | Cover crop | NA | NA | NA | NA | NA | NA |
| Year total | | | **322** | **253** | **134** | **77.6** | **0.53** |

Similarly, the peak diurnal $LE$ flux for soybean was larger than that of winter wheat cover crop. The peak midday $LE$ for soybean varied between 250 and 650 W m$^{-2}$ and averaged 450 W m$^{-2}$. The timing of the daily $LE$ peak coincided with maximum $R_n$, typically occurring between 12:00 and 13:00 local time. The diurnal $H$ flux was nearly symmetrical around noon, reaching maximum between 13:00 and 14:00, often reaching zero by 17:00, and remained below zero during the night (Figure 3). Latent heat flux increased throughout the summer season, reaching maximum in late July. Across the years, the season average, $G$, varied between 13.7–41.8 W m$^{-2}$ day$^{-1}$ and reached maximum value during spring fallow (i.e., before soybean planting season).

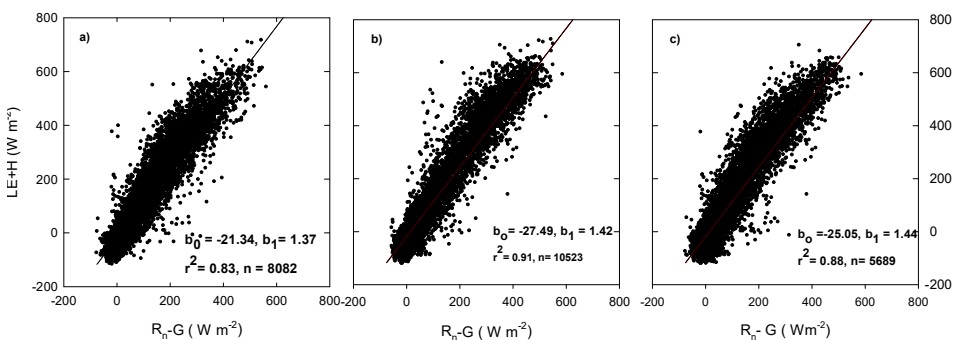

**Figure 3.** Energy closure at Winfred Thomas Agricultural Research Station for the measurement years of 2007 (**a**), 2008 (**b**), and 2009 (**c**).

Seasonal variation in $H$ and $LE$ across the measurement years occurred despite that $R_n$ rates were relatively consistent, suggesting that precipitation and crop phenology were major determinants of energy partitioning (Figure 2). The fraction of $H$ to total $R_n$ changed with seasons, it increased in February and began to decline in late April reaching a minimum in July. In general, pronounced $H$ flux relative to $LE$ was observed during drier years, non-growing and winter seasons. Across seasons, both the winter wheat cover crop and soybean partitioned more of the available energy into $LE$ than $H$, as evidenced by low Bowen ratio values. Soybean canopy partitioned more $R_n$ into $LE$ (76%) than winter

wheat cover crop. In 2009, soybean partitioned significantly higher available energy into *LE* relative to *H* than any other year.

### 3.3. Energy Closure

　　We examined the energy balance closure over the entire year and monthly to examine temporal trends of energy components on energy balance and crop type on closure (Figure 3). In the annual totals for 2007, 2008, and 2009, (*LE* + *H*) accounted for about 0.83, 0.81, and 0.93 of ($R_n$–*G*), respectively. Lack of closure became more evident when the dataset was analyzed by month, a proxy for crop phenology, with an apparent increase in the trend of monthly energy balance closure from winter wheat cover crop to the soybean cropping periods. Averaged across years, closure slightly improved (+3%) overall and energy balance closure was evident (i.e., 67% in the summer cropping season vs. 64% over the winter cropping season).

### 3.4. Daily and Seasonal Course of Albedo

　　Initially, when the 30 min time series data were plotted for the 24 h period, the influence of the crop on albedo did not clearly show the expected diurnal trend. This was in part because of the compounding effect of the angle of elevation of the sun on albedo. Thus, trends were analyzed for shorter time periods, and we focused on a 5 h period , i.e., between 09:00 h and 14:00 h local standard time (LST), during which maximum heat exchanges occur at the site. Variation in the diurnal trends of albedo—the ratio of net radiation at the surface over incoming shortwave radiation ($R_n/S_d$)—closely followed the time of day (a proxy for sun angle), while at a seasonal scale it was related to crop type (Table 2, Figure 4).

　　Albedo was relatively constant across the growing and off seasons, ranging between 0.14 and 0.18 across all measured years (Table 3). Typically, reflectivity was enhanced by, and strongly related to, low solar elevation angles, as evidenced by the increased albedo values in Figure 4.

　　At a seasonal scale, crop phenology and surface soil moisture status strongly regulated albedo. In all cases, albedo during the crop-growing seasons was lower than that observed during the dormant season. The summertime albedo tended to be lower than winter months, apart from 2007, which was (0.18) comparable to the other years. The winter albedo reached maximum, as high as 0.29, partially caused by a high reflectance of the winter wheat cover crop with ranges from 0.14 to 0.18 during the fallow period, from 0.15 to 0.18 winter wheat crop, and from 0.17 to 0.18 the for the soybean crop. The diurnal amplitude and magnitudes of albedo were slightly greater for winter wheat than soybean crops, indicative of the highly reflective nature of the smoother surface of the grass and inefficient radiation trapping by the winter wheat canopy. During the fallow period, $R_n$ is smaller than when crops are present on the ground, in part, because outgoing long-wave radiation fluxes are larger.

**Table 2.** Seasonal minimum, average, and maximum albedo values.

| Year | Season/Crop | Albedo | | |
|---|---|---|---|---|
| | | Minimum | Average | Maximum |
| 2007 | Non-growing | 0.12 | 0.18 | 0.22 |
| | Winter wheat | 0.10 | 0.18 | 0.29 |
| | Soybean | 0.12 | 0.18 | 0.22 |
| 2008 | Non-growing | 0.10 | 0.14 | 0.21 |
| | Winter wheat | 0.10 | 0.16 | 0.25 |
| | Soybean | 0.13 | 0.17 | 0.23 |
| 2009 | Non-growing | 0.11 | 0.18 | 0.18 |
| | Winter wheat | 0.10 | 0.15 | 0.29 |
| | Soybean | 0.10 | 0.17 | 0.27 |

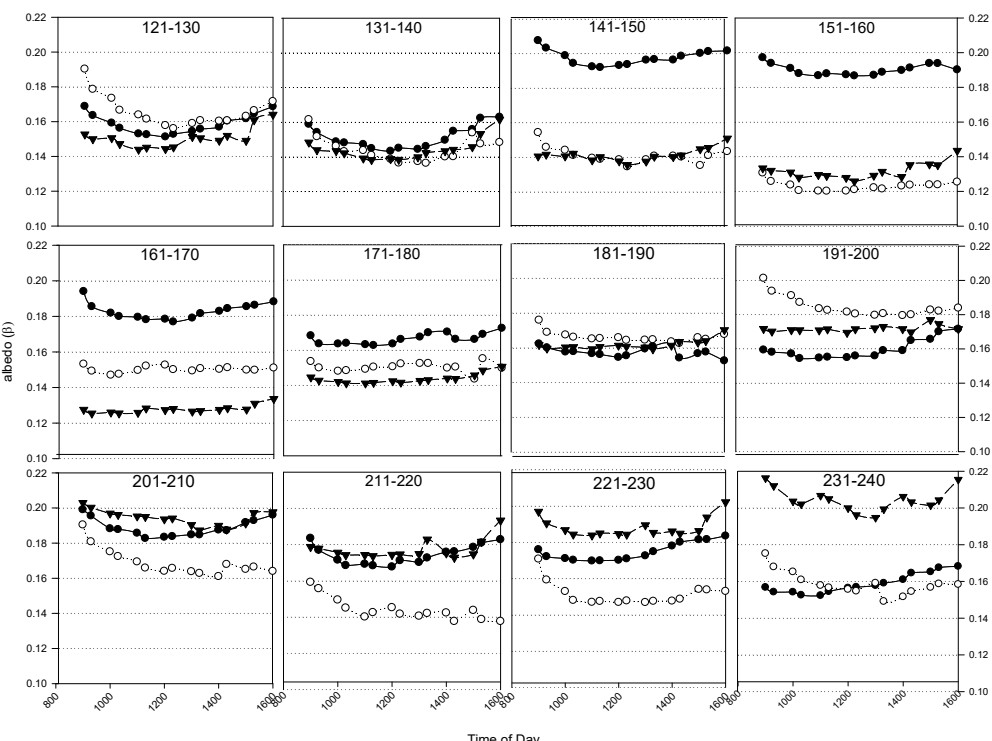

**Figure 4.** Ten-day binned average course of albedo for DOY 121–240 between 9:00 AM to 16:00 PM (LST) hours during 2007 (black circle, filled), 2008 (white circle), and 2009 (filled inverted triangle) mornings and in late afternoons. The midday average 0.16 of albedo for winter wheat cover crop were lower (range 0.10–0.22) than for soybean (0.17).

**Table 3.** Seasonal (growing and non-growing) minimum, average, and maximum values of measured *ET* (mm), precipitation (mm), and ET/P ratio (evaporative fraction) at WTARS for the period 2007–2009.

| | | *ET* (mm) | | | | *P* (mm) | *ET/P* |
|---|---|---|---|---|---|---|---|
| **Year** | **Season** | **Min** | **Max** | **Mean** | **Total** | **Total** | |
| 2007 | Non-growing | | | | 139.4 | 113.1 | 1.23 |
| | Soybean | 0.23 | 3.44 | 1.33 | 220.0 | 259.0 | 0.85 |
| | Winter wheat | 0.24 | 4.80 | 2.47 | 134.0 | 112.0 | 1.20 |
| | cover crop | 0.07 | 3.73 | 1.70 | | | |
| Year total | | | | | **493.4** | **567** | **0.87** |
| 2008 | Non-growing | | | | 178.8 | 294 | 0.61 |
| | Soybean | 0.16 | 3.90 | 1.82 | 260.0 | 304.0 | 0.86 |
| | Winter wheat cover crop | 0.20 | 4.90 | 2.83 | 304.0 | 682.0 | 0.45 |
| | Non-growing | 0.15 | 4.57 | 1.74 | | | |
| Year total | | | | | **743.0** | **1280** | **0.58** |
| 2009 | Non-growing | | | | 80.5 | 34.5 | 2.33 |
| | Soybean | 0.19 | 4.18 | 2.51 | 282.0 | 494.5 | 0.57 |
| | Winter wheat cover crop | 0.16 | 4.48 | 1.82 | 383.0 | 827.0 | 0.46 |
| | Non-growing | 0.18 | 7.50 | 2.44 | | | |
| Year total | | | | | **746** | **1356** | **0.55** |

### 3.5. Evapotranspiration

We found large differences in total *ET* among years with clear seasonal patterns (Figure 5). On an annual basis, year-long cumulative *ET* of 493, 743, and 746 mm during 2007, 2008, and 2009 resulted in the highest mean *ET/P* ratio of 0.87, in 2007, followed by

0.58 in 2008 and 0.55 in 2009. The average annual soybean *ET* from 2007–2009 was 254 mm and ranged from 220–282 mm, with the lowest amount being in the summer of 2007. During the early g8 rowing stages, the seasonal course of *ET* exhibited different patterns each year of the study. In 2007, the peak value of *ET* was observed in July but was not sustained due to drought.

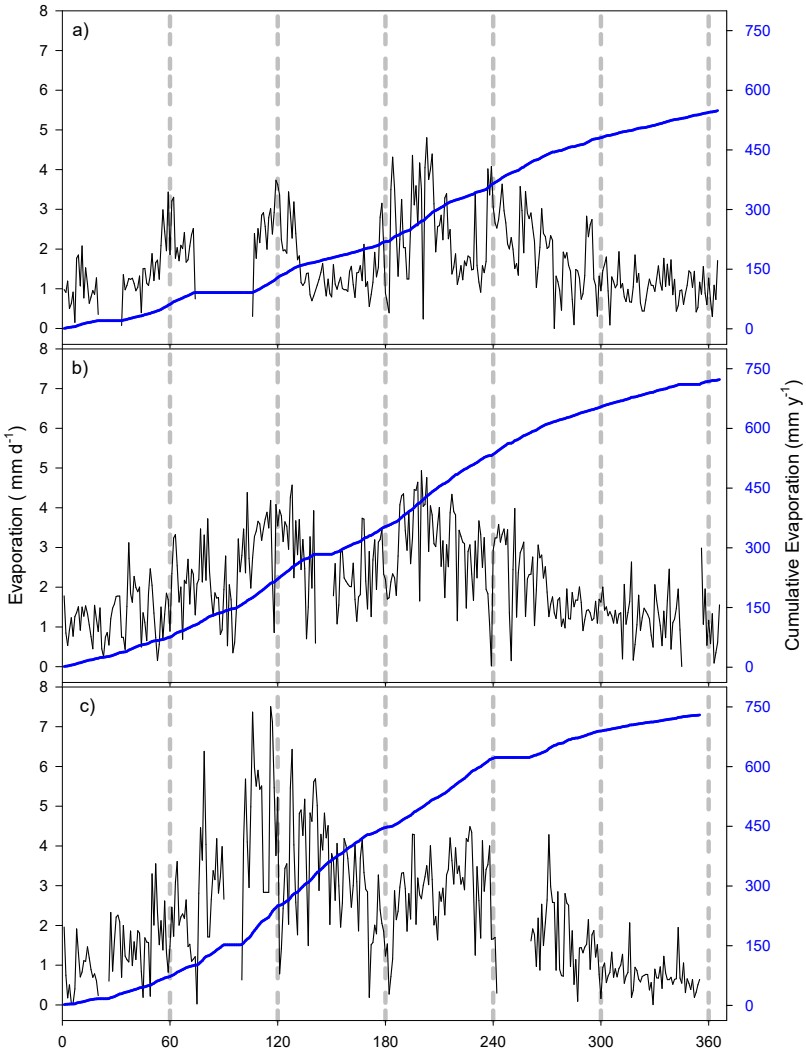

**Figure 5.** Daily mean values of evaporative flux for (**a**) 2007, (**b**) 2008, and (**c**) 2009. The blue diagonal line is the year cumulative annual of *ET* 493, 743, and 746 mm during 2007, 2008, and 2009, respectively.

The peak rates of *ET* for soybean ranged from 3.7 mm d$^{-1}$ in 2007 to 4.9 mm d$^{-1}$ in 2008 and averaged about 4.7 mm d$^{-1}$. On average, *ET* of soybean was 50–56% higher than that measured from the winter cover crop. In 2008 and 2009, peak *ET* was observed in August, when the *LAI* was at its annual maximum. On average, daytime ($R_S > 10$ W m$^{-2}$) sums of *ET* were near zero in winter in both 2007 and 2008 and showed increased rates from February to August. The seasonal pattern of *H* was related to the difference between incoming net radiation and the *LE* flux. In 2007, the below-average precipitation caused near-surface (5–10 cm) *SWC* to be extremely low, with values declining steadily from 0.16 m$^3$ m$^{-3}$ at the beginning of 2007 to 0.04 m$^3$ m$^{-3}$ by midsummer. Annual *ET* in 2007 was ~34% lower than both 2008 and 2009. Extended hot periods in early July, August, and September in 2007 increased *ET*, with a concomitant increase in *LE*. *ET* then began to decrease in September, before canopy senescence was visible, which caused *H* to remain

steady during this period, despite declining $R_n$. After the soybean harvest, *ET* continued to decline in October and November, reaching a low rate in December.

Changes in the measured *ET* are also evident over the course of the growing season (Figure 5). The seasonal variation in *ET* corresponded closely with precipitation and the associated changes in available soil moisture. The mean daily *ET* during the winter wheat growth period ranged from 1.7 to 2.5 mm d$^{-1}$, when the cover crop was at maximum growth. Daily *ET* of soybean ranged from 0.19 to 4.90 mm d$^{-1}$ (Table 3). Shortly following soybean planting, direct soil evaporation dominated *ET*, but its relative contribution diminished with the growing season (i.e., with increasing *LAI*). The highest values of daily *ET* (~2.8–3.6 mm d$^{-1}$) occurred during summer seasons, especially in August, when soybean achieved its highest *LAI* with cumulative *ET* values of 220, 260, and 282 mm during the 2007–2009 soybean growing seasons. The *ET* from bare soil had peak rates that ranged between 2.6 and 4.2 mm d$^{-1}$.

## 4. Discussion

### 4.1. Effects of Seasonal Changes in Radiation and Water Availability on Partitioning of Available Energy and ET

This study focused on highlighting the importance of seasonal precipitation in controlling both energy partitioning and the magnitude of *ET* in the SE. Furthermore, our observation underscores the importance and the need for quantitative in situ measurements in examining and assessing energy and *ET* in the context of changing precipitation pattern. Several reasons have been put forward for variations in precipitation in the past several decades over much of the SE United States [10]. Previous studies have shown that agricultural productivity and surface energy balance may be sensitive to year-to-year variation in precipitation and warming [22]. Our study also shows that precipitation was the major driver for the observed variations in *LE*, *H*, and *G* during the observation years. During the summer, *LE* flux exceeded *H* in 2008 and 2009, but not during the driest year of 2007, indicating the sensitivity of evaporative flux to soil moisture depletion, despite nearly similar $R_n$ values across the years (Figure 2). Conversely, Burba and Verma, 2005 [23] found that differences in ET between ecosystems and the corresponding interannual variability were related mostly to soil moisture stress and variations in green foliage area. Of particular importance, our study allowed us to gain an understanding of the impacts of droughts on energy flux patterns in the context of the impact of a changing hydroclimate on croplands energy exchange and ET in the SE. Interannual variability in precipitation has been a common occurrence across the SE in recent years, often characterized by extreme drought [6–8,10,24–28].

Notable was the estimated lowest 2007 ET, among the years with lowest evaporative fraction, and this was evidenced by a record number of *VPD* days with values > 3.5 KPa (Figure 1). In the non-drought years of 2008 and 2009, above average precipitation, which led to higher *LAI* and lower ground heat flux, contributed to higher proportion of net radiation being available for latent heat along the transition from winter wheat to soybean canopy. Seasonal changes in *LE* fluxes were also more closely related to canopy development, as the highest *LE* (253 W m$^{-2}$) was related to high occurrences of green leaf area in 2009 and the lowest *LE* was in 2007.

The increase in the Bowen ratio of the winter wheat canopy during winter growing season indicates that a higher proportion of the $R_n$ was used to warm the atmosphere via *H* than was used in evaporating water (i.e., *ET*) from the sparse canopy cover. Likewise, the higher fraction of $R_n$ allocated to *LE* in 2008 and 2009 was consistent with lower values of daytime β and provided further evidence for the presence of readily available soil moisture in both years. Clearly, months with higher precipitation (i.e., proxy for growing seasons) had enhanced *LE* and reduced *H*. This observation is based on our results from the summer of 2007, when it was shown that the impact of moisture sensitivity of *LE*, although much stronger in the other years, was not as severe as expected. In studies involving other cropping systems, similar trends have been observed where seasonal distribution of

precipitation and associated changes in soil–water dynamics significantly contributed to observed differences in daily and seasonal distribution of *LE* and *H* [29–31].

Across the years, the dominance of *H* in the energy budget was most pronounced at later stages, i.e., at physiological maturity or outside the growing season. For winter wheat, both *H* and *LE* exhibited linear dependency on $R_n$; however, this crop apportioned higher *H* relative to *LE*, in part, because of incomplete surface coverage (i.e., low *LAI* of the cover crop), compared to soybean. In part, this was because most of the precipitation during the non-growing season is rarely stored in the ground, with the excess water either quickly turned into surface runoff or evaporated from the surface. For example, the measured seasonal differences between *LE* and *H* were 79 and 152 W m$^{-2}$ for 2007 and 2008, respectively, a pattern that correlated well with precipitation amounts of 653 and 1153 mm y$^{-1}$ for the same years. As noted in the Gebremedhin's et al., 2012 paper [24], significant reduction in carbon loss during the summer and winter growing seasons of 2007, providing compelling evidence that changes in precipitation (both amount and duration) can influence surface–atmosphere $CO_2/H_2O$ exchange in the SE.

Biophysical variables alone, however, cannot explain differences in energy partitioning. In addition to physical forcing, sensitivities of energy partitioning can also covary with crop development within any cropping season by way of changes in leaf area. The $LE/R_n$ remained constant before May, and then increased with the developing soybean, reaching a maximum value in late July with about 0.53 when the *LAI* was approximately between 4 and 6 (data not shown). On the contrary, the lowest value of $H/R_n$ (~0.13) occurred in summer when most of the solar energy received by the ecosystem was consumed in *ET*. Winter wheat is typically planted in late October, but crop growth is very slow during the winter months (January to early March). Winter wheat greening begins in March and accelerated growth follows during the spring months of April and May. Similarly, albedo increased from early March through April, then progressively fell off as the crop reached maturity. Our results also showed that albedo increased with diminishing *LAI* (i.e., at physiological maturity and senesces)—as expected—resulting in maximum values of 0.16 values, attributed, in part, to soil reflectance and soil moisture availability.

This difference in sensitivity could also be explained by whether variation in *SWC* was in sync with other potential drivers of *LE*, e.g., crop growth stage, *LAI*, or *VPD*. Other studies have also confirmed that strong and positive responses of partitioning between *LE* and *H* were not only functionally related to physical factors but also to plant-specific attributes, such as canopy structure, *LAI*, physiology (i.e., stomatal conductance) or agronomic practices (i.e., planting density, time of planting, or harvest time) [32–34]. Ground heat flux (*G*) during the non-growing season was always positive and larger, indicating that a larger fraction of the radiation reaches the ground than when crops are present. Thus, differences in the $G/R_n$ ratio across the growing season and during the non-growing seasons could partly be explained by differences in canopy shading. Similarly, ground heat flux as a proportion of $R_e$ showed a marked increase during the non-growing season but decreased as a function of *LAI* during the growing season. During the summer, for example, soybean with the greatest *LAI* attenuated much of the radiation before it reached the ground, compared to the winter wheat cover, keeping the soil profile relatively cooler in summer months. Thus, the degree to which *G* becomes more responsive and sensitive to soil moisture also depends on whether the surface is partially (i.e., during drought) or fully covered by foliage. Among the energy components, *G* was the lowest of all fractions (Table 1, Figure 3). Daily *G* varied greatly over the seasons, with rapid increases in October and May (with no vegetation cover) and reaching its minimum during the summer.

Albedo is a variable that could change drastically over short period, since it responds to dynamic biophysical drivers and plant phenology and includes factors, such as diurnally changing cloud cover conditions, canopy structural changes in response to growing season, such as *LAI*, plant architecture, or morphology. Mean daily albedo varied with a diurnal trend of highs/lows during morning and later afternoon hours, respectively (Figure 4), a

pattern that seemed to be driven primarily by strong diurnal changes in sun angle. The difference between winter wheat and soybean albedo was small, compared with bare ground, albedo was the highest fallow month (data not shown). For example, substantial differences in reflected shortwave radiation were observed, with the highest and lowest albedo in 2007 and 2009, respectively, likely related to differences in precipitation and accompanied soil moisture conditions (Figure 1, Table 2). This suggests that on longer timescales (e.g., annual), variation in albedo is tightly linked to differences in precipitation than canopy properties.

Several studies have shown that diurnal trends of albedo corresponded strongly with the sun angle [12,34]. Although albedo for croplands is thought to be higher than other ecosystems (e.g., boreal and temperate forests), we did not find albedo to be consistently higher at our site. In fact, at this agricultural site, albedo was typically lower to intermediate (ref. Table 2), compared to other ecosystems. This pattern was largely aligned with results from similar observations, in which differences in albedo could not be explained solely by plant factors (e.g., phenology) without considering several interrelated biophysical factors [35–39].

### 4.2. Biophysical Controls on ET during Growing and Non-Growing Seasons

As a key component of the terrestrial water and energy budget, the *ET* process returns an estimated 70% of precipitation to the atmosphere [39] exerting a significant influence on the local and regional climate [40].

We observed similar patterns in annual cycles of radiation components, indicating that energy partitioning was primarily regulated by *P*, thus, the seasonal distribution of *ET* was greatly determined by the amount of *P* and the associated changes in *SWC* and crop type. Interestingly, previous studies have also reported that microenvironment and vegetation characteristics (e.g., *LAI* and crop development stage) were the two most important determinant factors interannual *ET* variation in agricultural systems. The inter-annual difference in *ET* was largely attributable to differences in precipitation. For example, in 2009, cumulative *ET* ranged from 100–172 mm, and this contributed 16–28% of the total annual precipitation. The higher $R_n$ to $H$ ratio in 2007 indicates that most of the precipitation was evaporated into the atmosphere—primarily from soil evaporation—because of sparse crop surface coverage. In contrast, 58% of the annual precipitation was converted to *ET* in the form of crop transpiration in 2009. Averaged across the years, about 59% of precipitation was returned through *ET* on an annual basis for the three years in this soybean–winter wheat cropping system. The ratio of actual *ET* to precipitation was lower than that previously reported for rainfed soybean [28,29,41] and northern temperate grassland [42], but it was comparable to wet temperate grassland in Japan [37].

In our study, daily *ET* values during the winter wheat growing seasons (November–April) ranged from 0.02–7.50 mm d$^{-1}$ (mean 1.96 mm d$^{-1}$), with season-long cumulative *ET* values of 134, 304, and 379 mm during the 2007–2009 growing seasons, suggesting a severe drought and warmer year. In general, lower *ET* rates of about 0–5 mm day$^{-1}$ were observed during the winter wheat growing season. This mid-winter rate was comparable to or somewhat lower than expectations for well-watered summer crops, the low rates of winter *ET*, and the observation that dormant-season *ET* was largely sensitive to the ample *SWC*. As expected, *ET* observations over the winter wheat canopy were substantially lower and exhibited less-pronounced variation over the growing season than those measured over soybean canopy, where *ET* ranged from 3.18–0.67 d$^{-1}$ to 3.96–0.65 mm d$^{-1}$. The midsummer rate of daily *ET* was 3–4 mm d$^{-1}$, which is comparable to or somewhat lower than expectations for well-watered, upland grassland [37,42].

Elevated summer temperatures also increased the drought vulnerability of the soybean canopy and induced significant *ET* loss, altering the partitioning of energy by reducing *LE* (i.e., increasing *H*). We note, however, that the winter wheat cover crop growing season is longer by an average of 2–3 months than the summer soybean season. The comparatively lower rates of *ET* in 2007 are tightly correlated with that of limited *SWC*. On average,

mean $ET$ at the site was in the range of 0.16–4.90 mm d$^{-1}$ during soybean season and 0.07–7.50 mm d$^{-1}$ (Table 3). In comparison with growing seasons, $ET/P$ for bare soil was consistently higher, with values almost three times higher than during growing seasons. The peak rates for $ET$ from non-cropping season ranged between 0.16–4.2 mm d$^{-1}$ across the years, independent of $R_n$. It is likely that the decrease was due to the absence of vegetation cover and the associated reduction in roughness heights. During the transition to the growing season, a shift in energy dissipation dominated by $H$ was apparent.

As noted earlier, the soybean $ET$ rates we measured in the summer of 2007 were significantly lower than the following consecutive observation years. During the severe drought year of 2007, the annual $ET$ and $P$ (220 mm, 259 mm, respectively) fell in the lower range of $E$ and P (163 to 481 mm) reported for rainfed soybean in other climatic zones [40,43]. The $ET/P$ ratios of 0.85, 0.86, 0.37 were also significantly higher than those reported for grassland [43] but comparable to $ET/P$ for Midwestern crops [41] and northern temperate grassland [44]. These differences in seasonality may help explain why ecosystems with low mean annual temperature but greater temperature variation often have a higher inter-seasonal $ET$ than warmer ecosystems with lower annual amplitudes in temperature.

## 5. Conclusions

Over the past two decades, most of the flux studies have occurred on forest and grassland systems with less attention on croplands. Evaluating the EC flux system performance in the turbulent atmosphere is an important objective criterion for assessing instruments response in sampling the expected frequency ranges of eddies carrying mass and energy. The purpose of the present study was to (i) provide quantitative assessment of seasonal and inter-annual variation in energy partitioning and $ET$, and (ii) better understand the functional relationships between energy flux and underlying biophysical factors across temporal scales. Three contrasting growing years, including a record drought year, have allowed us to examine the energy partitioning pattern and the sensitivity $ET$ to weather perturbations during the 2007–2009 growing seasons in humid subtropical Alabama climate. We conclude that (i) magnitude and pattern in energy flux partitioning over winter wheat–soybean cropping system were crop-specific, (ii) variation in precipitation explained a large portion of the observed variability in partitioning of $R_n$ into $H$, $LE$, and $G$, and (iii) shortening the length of fallow periods (between plantings) and presence of crop residues on soil surface are key factors in buffering $SWC$ and reducing $ET$, particularly during drought years.

Agricultural systems are often characterized by a suite of cropping and soil management practices, resulting in significant challenges for long-term measurement of surface energy fluxes. Our results underscore the need, in the decades ahead, for a coordinated regional network that expands existing monitoring programs under the USDA's Long-Term Agroecosystem Research (LTAR) Network program into understudied regions, one that makes use of standardized sampling methods. Such a network is destined to become a critical tool for planning new and adaptive farming practices in response to future climate changes.

**Author Contributions:** Conceptualization, EC field data collection, flux data analysis, original draft preparation; M.G. Review and revising; J.B. and I.R. All authors have read and agreed to the published version of the manuscript.

**Funding:** M. Gebremedhin: a doctoral student at the time, was supported by the Department of Natural Resources and Environmental Sciences at Alabama Agricultural and Mechanical University, Huntsville, Alabama.

**Data Availability Statement:** Data can be provided upon request. E-mail Maheteme.gebremedhin@kysu.edu.

**Acknowledgments:** The authors are grateful to Winfred Thomas Agricultural Research Station's staff at AAMU for operational support. We would also like to acknowledge the comments and feedback received from the anonymous reviewers.

**Conflicts of Interest:** The authors declare no conflict of interest.

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
