# Peer review of "Soil Management and Microclimate Effects on Ecosystem Evapotranspiration of Winter Wheat–Soybean Cropping in Northern Alabama"

_atmosphere, doi:10.3390/atmos13101653_

Round 1

Reviewer 1 Report

This manuscript cannot be published in Atmosphere. It is plenty of errors of calculations. Moreover, it does not cover  the objectives to be addressed.

A few concerns to be addressed;

Line 26. ‘…dimensions …’. Please, reword.

Lines 25 to 125. It is too long. It requires a drastic reduction to focus a reader on the problem to be addressed.

Lines 125 to 141. This is very confusing. What are exactly the objective/s to be addressed?

Line 145. Rewrite ‘….three-dimensional wind speed ….

Line 155. Rewrite ‘..sensible heat flux  and latent heat flux

Line 159. Rewrite’.. sampled the turbulence… or much better …operated at 10Hz..

Lines 161 and up. The following details are not necessary in this context because the deployment height was at 3 m. Please, delete it

’ ..The CSAT and IRGA measurements 161 were made within the well-mixed surface layer, i.e., zh > 5zm + d, where zh is the measure- 162 ment height (m), zm is the surface roughness length (m), and d the zero-plane displacement 163 height (m) (the theoretical height above the ground where the logarithmic wind profile 164 vanishes to zero). The measurement height zh was 10 times the average zh of the winter 165 wheat canopy (0.30 m, at peak developmental stage). For soybean zh was nearly 7.5 times 166 higher than the average d of soybean (0.4 m) with roughness length approximately 0.06 167 m.’

Lines 170. This must be deleted. Why do you need the post-processing EC details?

Prior to 170 estimating surface flux, wind velocities were rotated so that the instrument coordinate 171 systems were aligned with the local mean streamlines [42]. Hence, the mean x-axis was 172 mathematically rotated in order to make it parallel to the local streamline and the z-axis 173 orthogonal to the x-axis [43]. Before calculating the final turbulent flux, a density correc- 174 tion was added to account for temperature and heat fluctuations [44].  

Lines 201. Please, correct u* .

Line 211. This is obvious, delete. ‘…We adopted the sign convention that when Rn is positive it 211 implies the surface is gaining energy

Eqs 4 , 5 and 6 are they correct?.

Line 234. G was directly measured 234 using a soil heat plate at 10 cm depth?. What happens with the storage?

Author Response

Please see attached our response to reviewer 1.

Maheteme

Reviewer 2 Report

The submitted MS dedicates into an understanding of land-atmosphere energy and water fluxes over a cropland using eddy-covariance measurement. The topic is interesting and relevant for the journal. The manuscript was very nicely written and the experiment was well designed. The processing of EC data is acceptable. The figures are neat. Overall, I suggest “Minor Revision” for the MS.

Comments:

Line 134: I suggest to add “over a typical cropland” after “drive surface-atmosphere flux” here.

Line 136 and 137 (The second and third objectives): I am confused that why “forested” and “tree species” are used here? I see the MS is mainly about cropland with wheat and soybean.

Line 196-Line 223: I do not think these (and the Equation 1 and the Equations 3-5) are necessary for a scientific paper because these are basic knowledge. Did you use some software (e.g., EddyPro) to process the EC data? If so, just showing the software name is enough since these are standard processing.

Line 194: Please show how did you fill the gaps (~22% data) to further calculate the daily/seasonal mean energy fluxes.

Line 98: Consider to cite https://doi.org/10.1029/2019WR024867  This is an interesting large-scale ET study over United States with multiple ET datasets.

Table 1: For the energy flux, the unit is Watt per m2.  So you need to use “average” (rather than the “sum”) of 30-hourly EC-observed data to calculate the daily/seasonal/annual mean values. This is particularly true for the bottom line of each year where the so-called “Year Total” is shown. Please change to “average” here. Summing the energy fluxes is not correct.

Author Response

Please see attached our response to R2.

Reviewer 3 Report

The study authored by Gebremedhin et al. explores how to reduce evapotranspiration through cover crops. The study is based on detailed in situ observation for a period of 3 years on a single site in North Alabama, USA. The study is well designed and it worth being published in Atmosphere journal. The major weakness point consists in the fact that the data are quite old and, during a time of sudden climate change, the results could fade from their scientific relevance. The comments below are meant to improve the overall quality of the paper.

 Major comments:

The authors should explain why exactly the period 2007-2009 was used in this study and since these data are quite old what the current study brings as novelty.

The Introduction is too large and needs to be reduced only to strictly relevant information.

The title indicates North Alabama, the abstract refers to the entire US South-East but the study is developed based on single point measurement. I recommend to adapt the title and to restrict the idea on the abstract to this site.

The Discussion part lacks in-depth discussion of other studies and comparison with other previous studies.

Minor comments:

The authors should define “energy partitioning”.

L26: no comma needed after “including”

L30: The units should be given in an universal metrics (probably hectares of even better square km)

L35-36: Please rephrase. The ending of the phrase is not coherent with the rest.

L333: Please correct the time notation (13:00 or 1300)

Author Response

Please find attached our response to R3.

Maheteme

Round 2

Reviewer 2 Report

The authors have done a good job in revision. The MS is now acceptable.

Only comment: Ref 20 has a typo in the authors name, please recheck the refrence style again.

Author Response

Only comment: Ref 20 has a typo in the authors name, please recheck the reference style again

Thanks! The typo error is nor corrected.

Reviewer 3 Report

The authors responded adequately to all my comments. Even if I recommend the publication in the current form the authors are pleased to read carefully the manuscript one more time for minor aspects as commas.

Author Response

please see attached our response to Reviewer 3.